# Insight into the Mechanisms and Clinical Relevance of Antifungal Heteroresistance

**DOI:** 10.3390/jof11020143

**Published:** 2025-02-13

**Authors:** Yanyu Su, Yi Li, Qiaolian Yi, Yingchun Xu, Tianshu Sun, Yingxing Li

**Affiliations:** 1Department of Laboratory Medicine, State Key Laboratory of Complex Severe and Rare Diseases, Peking Union Medical College Hospital, Chinese Academy of Medical Science and Peking Union Medical College, Beijing 100730, China; yanyu_s_10023@yeah.net (Y.S.); liyi67@pumch.cn (Y.L.); yqlotus@163.com (Q.Y.); xycpumch@139.com (Y.X.); 2Beijing Key Laboratory for Mechanisms Research and Precision Diagnosis of Invasive Fungal Diseases (BZ0447), Beijing 100730, China; 3Graduate School, Chinese Academy of Medical Science and Peking Union Medical College, Beijing 100730, China; 4Clinical Biobank, Center for Biomedical Technology, National Science and Technology Key Infrastructure on Translational Medicine, Peking Union Medical College Hospital, Chinese Academy of Medical Sciences and Peking Union Medical College, Beijing 100730, China; 5State Key Laboratory of Complex, Severe, and Rare Diseases, Chinese Academy of Medical Science and Peking Union Medical College, Beijing 100730, China; 6Biomedical Engineering Facility of National Infrastructures for Translational Medicine, Institute of Clinical Medicine, Peking Union Medical College Hospital, Chinese Academy of Medical Sciences and Peking Union Medical College, Beijing 100730, China

**Keywords:** antifungals, drug resistance, heteroresistance, aneuploidy, Copy Number Variations (CNVs)

## Abstract

Antifungal resistance poses a critical global health threat, particularly in immuno-compromised patients. Beyond the traditional resistance mechanisms rooted in heritable and stable mutations, a distinct phenomenon known as heteroresistance has been identified, wherein a minority of resistant fungal cells coexist within a predominantly susceptible population. Heteroresistance may be induced by pharmacological factors or non-pharmacological agents. The reversible nature of it presents significant clinical challenges, as it can lead to undetected resistance during standard susceptibility testing. As heteroresistance allows fungal pathogens to survive antifungal treatment, this adaptive strategy often leads to treatment failure and recurring infection. Though extensively studied in bacteria, limited research has explored its occurrence in fungi. This review summarizes the current findings on antifungal heteroresistance mechanisms, highlighting the clinical implications of fungal heteroresistance and the pressing need for deeper mechanism insights. We aim to bring together the latest research advances in the field of antifungal heteroresistance, summarizing in detail its known characteristics, inducing factors, molecular mechanisms, and clinical significance, and describing the similarities and differences between heteroresistance, tolerance and persistence. Further research is needed to understand this phenomenon and develop more effective antifungal therapies to combat fungal infections.

## 1. Introduction

Fungal infections pose a significant and growing threat to public health worldwide, particularly in immunocompromised patients [1,2]. These infections are associated with high mortality rates, especially in vulnerable populations, such as neonates, those with HIV/AIDS, cancer patients undergoing chemotherapy, and transplant recipients [3,4,5,6,7]. For instance, invasive fungal infections have mortality rates ranging from 40% to 70% in candidiasis, 40% to 80% in cryptococcosis and can reach over 90% in aspergillosis, depending on populations and infection types [8,9,10].The economic cost of fungal infections is substantial, exceeding billions of dollars annually due to prolonged hospital stays, the expense of antifungal therapies, and complex management strategies [11,12,13]. Moreover, the increasing prevalence of antifungal-resistant infections complicates treatment, making fungal infection a major global health challenge [14]. Traditionally, antifungal resistance has been considered as being due to inherent and stable genetic mutations, with elevated MIC (Minimum Inhibitory Concentration) values, that diminish drug efficacy. However, recent research has uncovered a more complex and dynamic form of resistance: heteroresistance. Fungal heteroresistance, in which a small subpopulation of a fungal strain exhibits resistance to antifungal drugs while the majority remain susceptible, presents a significant global threat. This phenomenon complicates treatment, as it can lead to treatment failure despite the apparent efficacy of the drug against most of the fungal cells, further exacerbating the public health crisis caused by fungal pathogens.

Heteroresistance was first discovered in 1947 in *Haemophilus influenzae* [15]. In bacteriology, it refers to the presence of a heterogeneous population composed of one or several subpopulations with increased levels of antibiotic resistance compared with the major population. This can be due to clonal heterogeneity during co-infection by colonies with different levels of resistance, or microevolution of a single colony [16,17]. The heteroresistance phenotype can either be sufficiently stable or not in rapidly reverting to a susceptible phenotype, or show instability that may affect laboratory test results [18,19]. Under persistent selective pressure, it may even evolve heritable properties, causing higher rate of treatment failure, and create an antibiotic resistance reservoir [20]. As for mechanisms, they vary among different pathogens in response to various kinds of drugs. For example, *Pseudomonas aeruginosa* shows heteroresistance to imipenem due to biofilm formation and *OprD* gene mutation [21], while heteroresistance to levofloxacin is linked to elevated expression of genes involved in DNA replication and repair [22]. Research on *H. influenzae* showed that imipenem heteroresistance was linked to alteration of PBP3 (penicillin-binding protein 3), drug influx and efflux changes [23]. It can also be induced by antibiotic exposure, through upregulation of stress-related pathways, etc. [24,25].

While extensively studied in bacteria, research on fungal heteroresistance remains comparatively limited. From a clinical perspective, heteroresistance has been implicated in treatment failure in murine models [26], but clinically important mechanisms have not been studied in sufficient depth. However, the clinical impact of heteroresistance leading to drug resistance during treatment should not be underestimated, as it may lead to prophylaxis failure, recurrent infection or relapse [3,27,28,29,30]. In this review, we summarize current findings to outline the characteristics, mechanisms, and clinical prevalence of heteroresistance in fungal pathogens, with the goal of providing a foundation for future investigations.

## 2. Methodology

This narrative review was undertaken to provide a comprehensive overview of antifungal heteroresistance, focusing on its molecular mechanisms and clinical implications. Relevant literature was identified through searches in PubMed and Web of Science using the keywords “antifungal heteroresistance”, “antifungal resistance”, “antifungal tolerance”, and “antifungal persistence”. Articles published from 1995 to 2025 were considered, with a focus on studies investigating genetic, epigenetic, and phenotypic mechanisms underlying heteroresistance, as well as its clinical outcomes and treatment methods. Priority was given to peer-reviewed research articles, reviews, and case reports relevant to clinical microbiology and antifungal therapy. Non-English articles, conference abstracts, and studies lacking direct relevance to antifungal tolerance/persistence/heteroresistance, or expressing the definition ambiguously, were excluded.

## 3. Antifungal Resistance Mechanisms

Fungi are significant eukaryotic pathogens in humans capable of causing infections ranging from superficial to life-threatening systemic diseases [8,31,32,33,34,35,36,37,38,39,40]. Their pathogenicity stems from immune evasion [41,42,43,44,45,46,47,48,49,50,51], adaptation to changing pH conditions and nutrient availability [45,47,52,53,54,55,56,57,58,59,60,61,62,63,64,65,66,67], tissue invasions [40,42,43,44,68,69], etc. These characteristics make fungal infections particularly challenging to treat, especially in immunocompromised individuals [4,7,70,71,72,73]. Currently, there are four main classes of drugs for treatment of fungal infections: azoles, echinocandins, polyenes (Amphotericin B, AmB), and 5-fluorocytosine (5-FC). Azoles act on fungal 14α-demethylases and inhibit ergosterol biosynthesis, producing toxic methyl sterols and leading to alterations in the permeability and metabolic state of fungal cells, which may result in growth inhibition or cell death [74,75]. Echinocandins target and inhibit β-1,3-glucan synthase, thereby inhibiting β-1,3-glucan production and impairing fungal cells in maintaining their shape and resisting external stresses [76]. AmB targets membrane ergosterol, leading to pore formation, altered permeability, and reactive oxygen species (ROS) accumulation, all of which led to eventual fungal cell death [77,78,79]. 5-FC is a prodrug that is imported into cells by cytosine permease, which is encoded by the *FCY2* gene. Once inside the cell, 5-FC is converted to 5-fluorouracil (5-FU) through the action of cytosine deaminase and uracil phosphoribosyl transferase, both encoded by the *FCY1* gene, inhibiting DNA and RNA biosynthesis [80,81]. Specific modes of action of these antifungal drugs are illustrated Figure 1.

In recent years, a concerning surge in the isolation of antifungal resistant strains has raised a critical alarm for public health. Mechanisms associated with azole resistance mainly include mutations in azole targets, efflux pumps and biofilm formation, etc. [82,83,84]. In *Candida* spp., upregulation of the efflux transporter genes *CDR1*, *MDR1* and mutations in transcriptional factor TAC1B and in the genes *ERG11* and *ERG3* [82,84,85,86,87], which result in for example, primary sequence changes Y132F and R398I in *C. parapsilosis* lanosterol 14α-demethylase [85], have been associated with fluconazole resistance. A point mutation in *C. neoformans* sterol *14*α-demethylase 484S gives drug resistance to fluconazole only [88]. In *A. fumigatus*, mutations in the *cyp51A* gene encoding sterol 14α-demethylase and mutations in the ABC transporter gene *atrF* encode common azole resistance mechanisms [84,89].

Currently, resistance to echinocandins is mainly associated with mutations in the *FKS1/FKS2* genes [82,90,91]. Other molecular mechanisms associated with echinocandin resistance have also been reported. Yu et al. found that the *ADA2* gene of *Nakaseomyces glabrata* (formerly *Candida glabrata*) is associated with resistance to three classes of antifungals, as evidenced by Δ*ada2* knockout strains having significantly reduced MIC values [51]. Singh et al. found that deletion of seven genes, including *MOH1*, *GPH1*, *CDC6*, *TCB1/2*, *DOT6*, *MRPL11* and *SUI2*, increased levels of resistance to echinocandins, but considering their small effect on caspofungin resistance or the concurrent occurrence with *FKS2* mutations, it is more likely that they create a genetic background in which the *FKS2* mutations are adaptive and less harmful, rather than a direct cause of resistance [92].

5-FC resistance is highly correlated with its pharmacological mechanism, i.e., mutations in any one or more of the key enzyme genes of the pyrimidine salvage pathway, which may involve mutations in the genes encoding the cytosine permease *FCY2*, purine *FCY1* encoding the cytosine deaminase, *FUR1* encoding the uracil phosphoribosyl transferase, and *ADE17* [82,93]. However, other mechanisms have also been reported: Kannan et al. found that V668G substitution of a putative transcriptional activator (*MRR1*) led to the upregulation of *MFS7*, a multidrug transporter protein that mediated azole-5-FC cross-resistance in *Clavispora lusitaniae* (formerly *Candida lusitaniae*) [94] Billmyre et al. found that deletion of the mismatch repair gene, *MSH2*, in *C. deuterogattii* also led to an elevated rate of 5-FC resistance [95].

Fungal resistance to AmB is less common but has been reported in species such as *Candidazymus auris* (formerly *Candida auris*) and the *Candidazymus haemulonii* (formerly *Candida haemulonii*) complex [82,96]. Resistance to AmB in both species involves a variety of mechanisms, including alterations in cell membrane composition, cellular metabolic state, iron homeostasis, and ROS metabolism [58,77,78,96]. The molecular mechanisms of resistance to AmB in *C. haemulonii* were systematically discussed in the author’s previous review [40].

## 4. Heteroresistance, Tolerance and Persistence of Antifungal Drugs

At the cell population level, not all survival with fungicidal concentrations of antifungals can be attributed to elevated MICs. This is when heteroresistance, tolerance and persistence come into play. Here, we describe the differences between the three from the perspective of the performance of cell populations under drug stress, to minimize confusion about these concepts (Figure 2).

Heteroresistance, a concept first introduced for bacteria, is a form in which a minority subpopulation of resistant phenotype cells has a higher MIC, sometimes involving as few as one in one million cells, that coexists with a majority population of susceptible cells with a lower MIC [15]. Clinically, it refers to persistent infection, treatment failure, or relapse, despite in vitro susceptibility results indicating sensitivity [97,98]. It may be polyclonal or monoclonal. Polyclonal heteroresistance involves heterogeneous consortia with genetically distinct subpopulations or to the appearance of rare resistant mutants whose frequency increases with antibiotic exposure. Monoclonal heteroresistance is pure clones [18]. As a variant, it is more of continuous than binary (on/off). A study of heteroresistance properties showed that the incremental effects of multiple binary genetic switches, including on the efflux pumps *CDR1*, *PDH1*, *PDR1* and *SNQ1*, etc. produced a spectrum of heteroresistance rather than binary states, in *N. glabrata* [99]. From the perspective of population evolution, it can be seen as a strategy that helps the fungal population survive diverse environments [100]. In vitro, heteroresistance can be obtained through antifungal induction or non-antifungal agents and artificial selection, the last of which may produce the longest-lasting phenotype [57,101]. Repeated culture in stress-free environments can abolish heteroresistance [48]. Interestingly, this majority subpopulation can never be purified, meaning that a minority remains phenotypically resistant [102].

Tolerance is defined as an extension of the killing time, characterized by an increase in the MDK_99_ (Minimum Duration of Killing 99%) time [103]. It refers to the whole population that can survive a transient exposure to antifungals at otherwise lethal concentrations, without an increase in the antifungal MIC [104]. Furthermore, some 5–90% of the tolerant cells can slowly proliferate [105]. This definition applies to fungicides such as echinocandins and polyenes. However, for fungistatic drugs (e.g., azoles), tolerance has a different definition, i.e., the ability of a tolerant strain to overcome growth inhibition faster than a sensitive strain at drug concentrations above the MIC [106]. This phenomenon can be measured using broth microdilution or disk diffusion assays [98]. The reversibility of this property has been confirmed for several fungi species [107,108].

Persistence and persister cells have been extensively studied in bacteria [109]. For fungi, persistence refers to a subpopulation of genetically susceptible fungal cells that survives fungicidal concentrations of antifungals and may lead to the emergence of genetically resistant isolates [110,111,112]. This phenomenon has been observed in intra-macrophage *N. glabrata* cells, in which a subpopulation survives and gradually evolves heritable resistance mutations in *FKS* [110]. Sometimes “persistence” is used interchangeably with “tolerance”, as the MIC values of the fungal cell population do not change in both phenotypes. Importantly, persistence is characterized by a biphasic killing curve, whereas tolerance is not [109]. In other words, “tolerance” means that all survive and some may proliferate, whereas “persistence” means that some survive but none proliferates. Comparison of these concepts can be seen in Table 1.

## 5. Mechanisms of Antifungal Tolerance

Fungal tolerance to antifungals arises from a variety of molecular mechanisms. For example, azole tolerance in *Candida* populations is often characterized by the ability to survive and proliferate slowly at concentrations above the minimum inhibitory concentration (MIC). This tolerance is typically temperature-independent and frequently associated with aneuploidy, which may be lost upon temperature changes or ploidy recovery [114,115]. Similarly, in *C. neoformans*, brain glucose has been shown to induce AmB tolerance without affecting the MIC. This process is mediated by the zinc-finger transcription factor Mig1 and involves more complex mechanisms, such as the inhibition of ergosterol synthesis and the promotion of competing compounds that contribute to antifungal tolerance [116]. In *C. neoformans*, altered mitochondrial metabolism caused by cell ageing drives increased ergosterol synthesis and ABC transporter upregulation and leads to azole tolerance [117].

Alterations in cellular structure and metabolic levels also play a key role in antifungal tolerance. Cell wall remodeling contributes to echinocandin tolerance, with changes in β-glucan synthesis or increased production of compensatory components, like chitin, strengthening the cell wall and reducing drug efficacy [118,119]. In addition, activation of cellular stress response pathways, such as the heat shock protein (Hsp) and calcineurin pathways, has been shown to contribute to echinocandin tolerance in *C. albicans* [120,121]. Moreover, single-cell transcriptomics have shown that the ribosome assembly stress response supports survival under fluconazole exposure above the MIC, highlighting the importance of stress responses in antifungal tolerance [122]. Together, these mechanisms illustrate how fungi adapt to antifungal treatments through a combination of genetic, transcriptional, and physiological changes, making effective treatment more challenging.

## 6. Mechanisms of Antifungal Persistence

Persistence is always linked to cell dormancy, a state in which the cellular metabolism temporarily ceases [111,123,124]. As the fungal pathogen enters the host organism, it is often phagocytosed by immune cells, such as macrophages. Studies on pathogen–phagocyte interactions have revealed the inducing effect of the phagocytic intracellular environment on the formation of persisters [110]. Comparing the metabolism of planktonic cells and persisters affected by ROS or pH-related environmental stress suggested a possible role for cellular stress response in persistence [110]. Ke et al. constructed a mouse model of pulmonary infection and demonstrated that AmB-tolerant persisters are enriched in *C. neoformans* cells with high levels of the stationary-phase protein Sps1 and the metabolic marker ergothioneine [112]. This provides insight into the critical role of the *EGT* gene, responsible for the synthesis of ergothioneine, in regulating AmB susceptibility [112].

There are also several key factors that enable persisters. The formation of persister cells does not require biofilm formation, but biofilm-containing persister populations are more tolerant of oxidative stress and better resist the fungicidal effects of AmB [125]. Inhibiting biofilms helps eradicate *C. tropicalis* persister cells [126]. As antifungals can increase levels of ROS, persister cells might obtain stronger antioxidant capacity by upregulating enzymes, like superoxide dismutases (SODs) or alkyl hydroperoxide reductase 1, etc. [127]. Such mechanisms have been confirmed via SOD inhibition which reduces persistence levels [128]. Formation of persister cells should be considered important, as they may cause chronic or recurrent infections that complicate treatment.

## 7. Mechanisms of Antifungal Heteroresistance

The molecular mechanisms underlying fungal heteroresistance are diverse and involve a complex interplay of genetic, transcriptional, and physiological factors without always involving heritable change. Detailed mechanisms can be seen in Table 2. In many cases, heteroresistance arises via transient change that allows the fungus to survive antifungal stress without permanent genetic change. This adaptability complicates treatment strategies, as the resistant phenotype may not be stable and may fluctuate in response to environmental conditions.

### 7.1. Aneuploidy and Copy Number Variations (CNVs)

Aneuploidy is a common heritable mechanism responsible for the most widely studied form of antifungal heteroresistance [148,149,150]. The presence of an abnormal number of chromosomes plays a crucial role in fungal survival and evolution by providing rapid adaptability in response to environmental stress, such as antifungal drug exposure [151,152]. Yang et al. found that exposure to fluconazole at subinhibitory concentrations for a short period of time (48 h) may cause *C. neoformans* to acquire different aneuploid chromosomes that confer heteroresistance to fluconazole and cross-resistance to 5-FC [153]. Thus, exposure to one class of antifungal can promote adaptation to similar or different antifungal agents, highlighting the plasticity of the fungal genome and raising serious public health concerns. By increasing the copy number of chromosomes containing drug resistance genes, such as those encoding transcription factors, efflux pumps or involved in ergosterol biosynthesis, aneuploidy leads to resistance against multiple antifungals. including azoles and echinocandins [99,129,154]. Non-antifungal agents can also induce heteroresistance. Zhang et al. found that the endoplasmic reticulum stress chemo inducer Brefeldin A led to aneuploidy in *C. neoformans*: disomy in chromosome 1 led to cross-resistance to two classes of antifungal drugs, fluconazole and 5-FC, plus hypersensitivity to AmB [139]. By altering gene dosage or expression levels, aneuploidy allows fungi to quickly adjust to adverse conditions without permanent genetic mutations, promoting survival in fluctuating environments [99,129,145,152,154]. Importantly, its reversibility offers a dynamic mechanism for fungi to balance adaptation and stability, in the population and for individuals [151,155]. As a result, aneuploidy plays a pivotal role in both fungal adaptability and the evolution of drug resistance, making it a key factor in heteroresistance and its clinical effects.

#### 7.1.1. *C. albicans*

Mechanisms associated with azole heteroresistance in *C. albicans* are diverse. Under fluconazole exposure, *C. albicans* rapidly evolves CNVs and aneuploidy in vitro [135]. Associated resistance is mainly related to chr5, which has the *ERG11* and *TAC1* gene [129,130]. *TAC1* encodes a transcription regulator of ABC transporter genes located on chr3 [156]. Loss of chr5 can result in enhanced susceptibility to azoles [130], and can also lead to enhanced susceptibility to AmB and increased resistance to 5-FC. The latter effect is due to the location of the negative regulator (s) of anti 5-fluorocytosin on chr5, while the former needs clarification [130]. Harrison et al. reported the appearance of “trimeras”, connected cells composed of a mother, a daughter, and a granddaughter bud, after exposure to fluconazole [131]. The same morphology is not found in genetically resistant strains, suggesting the potential role of trimeras in heteroresistance. The trimeras produce progeny with different chromosome numbers, increasing the chances of developing heteroresistance [131].

*C. albicans*’ heteroresistance to echinocandins is related to chr2, as chr2 trisomy can be induced after exposure to caspofungin [132]. It is also related to chr5, as chr5 aneuploidy after caspofungin exposure gave cross-resistance to caspofungin, micafungin and anidulafungin [133]. Yang et al. found three negative regulators of echinocandin susceptibility on chr5. These are *CHT2*, encoding a glycosylphosphatidylinositol (GPI)-dependent chitinase, a covalently bound cell wall protein, *PGA4*, encoding a GPI-anchored cell surface 1,3-β-d-glucanosyltransferase, and *CSU51* encoding another putative GPI-anchored protein [132]. They also found positive regulators, i.e., *CNB1* encoding a regulatory subunit of calcineurin B, and *MID1* encoding a putative stretch-activated Ca^2+^ channel of the high-affinity calcium uptake system [118]. However, the genes involved and whether they act together in echinocandin heteroresistance is not clear [118].

#### 7.1.2. *N. glabrata*

*N. glabrata* was considered a haploid and an asexual organism for decades, but recent studies have reported the instability of clinical isolate genomes, mainly due to the frequent change in ploidy [157]. Ploidy variation might promote the rapid adaptation of *N. glabrata* to the changing environment and support the evolution of new traits, like heteroresistance. Ksiezopolska et al. found chrE aneuploidy contributes to heteroresistance after exposing clinical isolates to anidulafungin [134]. Heteroresistance due to environmental stress may remain even after the stress [134]. *N. glabrata* can also form “trimeras” on exposure to azoles. This might be related to heteroresistance, and its mechanistic basis needs further investigation [131].

#### 7.1.3. *C. parapsilosis*

In *C. parapsilosis*, multi-center studies have shown that heteroresistance facilitates breakthrough infections in immunocompromised patients and may cause prophylaxis failure [3]. However, unlike *C. neoformans*, exposure of *C. parapsilosis* to tunicamycin did not reveal any relationship between aneuploidy and heteroresistance [3,153]. Evidence of aneuploidy in the heteroresistance was obtained by Harrison et al., and showed that *C. parapsilosis* can form “trimeras” when exposed to azoles [131].

#### 7.1.4. *C. auris*

For highly resistant pathogen *C. auris*, Zhai et al. reported heteroresistance towards echinocandins, the first-line treatment for *C. auris* infection [3]. In vitro evolution during fluconazole exposure is relatively slow, compared to *C. albicans*, and mainly composed of a single nucleotide polymorphism (SNP), with aneuploidy a minority effect [135,136]. Due to its haploid genome, some SNP changes in *C. auris* may have immediate phenotypic effects [137].

#### 7.1.5. *C. neoformans* and *C. gattii*

For *Cryptococcus* spp., heteroresistance is a critical factor in the treatment failure of cryptococcosis and involves a subpopulation that appears under antifungal exposure and retains the potential to grow under continuous drug stress [98]. Because of the broad definition, many studies have reported “tolerance” as cryptococcal azole heteroresistance and vice versa [158]. For *C. neoformans* and *C. gattii*, the heteroresistance to fluconazole is intrinsic and increased resistance can be induced [19,102]. A comparative study showed that *C. gattii* showed a higher level of heteroresistance to fluconazole than *C. neoformans* [159]. The main mechanisms that could be related to *C. gattii* heteroresistance are copy number increase in chr10, chr9 and chr11 for the VGI genotype and copy number increase in chr3 for the VGII genotype [138]. For *C. neoformans*, heteroresistance can be reduced or inhibited by several environmental factors, including temperature, media type, growth phase, and cell age [147]. One of its mechanisms is associated with the multiple types of aneuploid daughter cells produced by titan cells [56,140,141]. A genomic analysis of clinical *C. neoformans* isolates found a high rate of aneuploidy in heteroresistant colonies and recurrent isolates, with a predominance of chr1 disomy [142]. Strains with chr1 disomy can also be isolated from mouse brain during treatment with fluconazole [160]. Ngamskulrungroj et al. demonstrated a high incidence of chr4 disomy, which may be related to *SEY1* (GTPase with a role in Endoplasmic Reticulum morphology), *GCS2* (ADP-ribosylation factor GTPase activating proteins) or *GLO3* (ADP-ribosylation factor GTPase activating proteins) genes [143]. If any of the *ERG11*, *SEY1*, *GCS2* or *GLO3* genes are relocated to chr3, then the frequency of chr3 disomy could increase [143,144]. This is due to the presence on chr1 of *ERG11*, which is the drug target of fluconazole, and *AFR1*, which encodes the drug efflux pump. However, the *AFR1* gene does not directly lead to heteroresistance [102]. Other research on *C. neoformans*’ ploidy found that exposure to inhibitory concentrations of fluconazole diminished budding and growth while permitting nuclear events. This resulted in populations with increased DNA, whose content grow better in the presence of fluconazole, increasing aneuploidy and survival [161]. The mechanisms of *C. neoformans* heteroresistance need to be defined to identify clinical therapies that will prevent and eliminate drug-resistant *Cryptococcus* subpopulations.

### 7.2. Alterations in Gene Expression

Efflux pumps, such as those belonging to the ABC transporter and major facilitator superfamily (MFS), play a crucial role in antifungal resistance by actively expelling drugs from the fungal cell. In heteroresistant subpopulations, the overexpression of efflux pumps can confer a temporary survival advantage. A genomic analysis of clinical *C. neoformans* strains found a high rate of aneuploidy in heteroresistant colonies and recurrent isolates, with a predominance of chr1 disomy and up-regulated activity of efflux pumps [142]. Sykes et al. obtained serial isolates from the same patient and found that overexpression of the efflux pump *PDR11* contributed greatly to the induced fluconazole heteroresistance [146]. Marr et al. demonstrated that the induction effect of fluconazole on heteroresistance involved elevation of mRNA levels for ABC superfamily *CDR* genes [145].

### 7.3. Environmental Stress Induction

In addition to specific antifungals inducing aneuploidy, non-pharmacological stressors can also impact antifungal heteroresistance and even cross-heteroresistance. Although the resistance phenotype is not stable due to the intrinsic instability of aneuploidy, it has been suggested that external inducing factors and genomic instability have vital roles in drug resistance in *C. neoformans*. For example it was found that *C. neoformans* heteroresistance was detected at 35 °C but not at 30 °C, indicating the importance of environmental temperature [19]. “Titanization” in *C. neoformans* also plays a role in the formation of aneuploidy. Gerstein et al. found that, during cryptococcosis treatment, newly emerged polyploid titan cells produced daughter cells that were more resistant to fluconazole and thus adapted to the host environment. Furthermore, a single titan mother cell produced multiple types of aneuploid daughter cells, which might contribute to the survivability of progeny under different environmental stresses [56]. This process was associated with intracellular ROS accumulation and mitochondrial responses [141]. For *C. neoformans*, heteroresistance can be reduced or inhibited by several environmental factors, including temperature, media type, growth phase, and the age of cells [147]. We speculate that stress response pathways contribute to the generation of antifungal heteroresistance. Bosch et al. demonstrated that environmental stress, such as the nitrogen limitation commonly encountered in its natural habitats, increases the resistance of *C. neoformans* to AmB and fluconazole, and increases the frequency of heterogeneous resistance to fluconazole [57]. For *A. fumigatus*, clinically, long-term itraconazole treatment may cause decreased susceptibility. Progressive itraconazole exposure in vitro can reproduce such phenomena, independent of the mutation in *cyp51A* gene [162]. This is called secondary resistance, suggesting the existence of heteroresistance. However, there are also studies showing the existence of intrinsic heteroresistance [48,102], and mechanisms of stress induction need to be further explored.

## 8. Clinical Relevance of Antifungal Heteroresistance

### 8.1. Outcomes of Antifungal Heteroresistance

Several studies have confirmed the increased emergence of heteroresistance in clinical, compared with environmental, isolates, supporting the induction effect of antifungal exposure [101,158]. Infections caused by heteroresistant strains increase risk to patients and make clinical treatment more difficult. One study demonstrated that Trimethoprim-Sulfamethoxazole heteroresistant *Staphylococcus aureus* strains gave near identical mortality as the resistant strains, indicating clinical consequences [163]. For fungi, a mouse model showed that heteroresistant *C. cryptococcus* isolates suppress the host immune response to enhance evasion, achieve higher fungal burden in different organs, and increase virulence [48,102]. As for high-level heteroresistant strains in brain-infected mice, if untreated their 20-day mortality rate can reach 100% [160]. Heteroresistant *N. glabrata* isolates can lead to higher fungal burden in mice kidneys, making it harder to thoroughly clear the pathogen and resulting in persistent infection [99].

In bacteria, heteroresistance may progress during antibiotic therapy, leading to changes in clinical tests and treatment failures and even creating an antimicrobial resistance reservoir [20,164]. Similarly, the transient nature of antifungal heteroresistance allows pathogenic fungi to survive conventional antifungal treatment, even when most of the population is inhibited. Prior to the onset of infection, this can facilitate breakthrough infection or result in prophylaxis failure [3]. For diagnosed fungal infections, it may lead to persistent infections that require longer or more aggressive treatment regimens [27,28,29,30]. Despite initial susceptibility to the drug, heteroresistant subpopulations can expand during therapy, partially due to antifungal exposure, characterized as the emergence of resistance-related genetic mutations, and clinically as relapse or progression of the infection [110,151,152,153]. To make things worse, antifungal heteroresistance can also contribute to cross-resistance between different classes of antifungal drugs. For example, *C. albicans* chr5 aneuploidy caused by caspofungin exposure can result in cross-resistance to caspofungin, micafungin and anidulafungin [133] and, in *C. neoformans*, chr1 disomy may confer cross-resistance to both azoles and 5-FC [139]. This makes treatment options even more restrictive.

### 8.2. Diagnosis of Antifungal Heteroresistance

Conventional antifungal susceptibility testing methods, such as broth microdilution and disk diffusion assays, may not detect heteroresistant subpopulations [165,166]. An in vivo experiment with *K. pneumoniae* strain *SWMUF35*-infected mice showed contradictory results compared to in vitro experiments, which classified the strain as amikacin susceptible [28]. These tests typically measure the MIC for the bulk population, potentially overlooking small subpopulations that can survive higher drug concentrations. This limitation arises because heteroresistant subpopulations may exist at low frequencies or exhibit resistance only under specific conditions, such as prolonged exposure to antifungal agents or within certain host environments [153,165,167,168]. As a result, infections caused by heteroresistant strains may be mistakenly classified as susceptible based on conventional testing. This misclassification can lead to the selection of antifungal therapies that are ineffective against resistant subpopulations, resulting in treatment failure, persistent infections, or even the emergence of fully resistant strains [99,165]. In clinical practice, this underscores the need for more sensitive diagnostic approaches to accurately detect and address heteroresistance, ensuring appropriate therapeutic strategies and improved patient outcomes.

For heteroresistance, the population analysis profile (PAP) assay is recommended as the gold standard [169]. PAP has been proved to efficiently test the heteroresistance to azoles and echinocandins in *C. albicans*, *C. haemulonii*, *C. tropicalis*, *N. glabrata*, *C. parapsilosis*, *C. neoformans* and *S. cerevisiae*, etc. [30]. Recently the novel Kirby–Bauer disk diffusion method demonstrated superior diagnostic performance for heteroresistance detection, though the PAP test remains the current standard method [97]. Single-cell assays can also be used in heteroresistance detection. These include techniques such as flow cytometry, single-cell RNA sequencing (scRNA-seq), and microfluidics-based assays, allowing for the analysis of gene expression, cell viability, and drug tolerance in individual fungal cells [170,171]. Though the protocols are lengthier, they may reveal a wide spectrum of adaptation mechanisms [30]. Time-kill assays, conventionally used to monitor the survival of fungal cells over time in the presence of an antifungal drug, can also be used to detect heteroresistance. By measuring the rate of cell death at various time points, this method can identify delayed growth or survival of heteroresistant subpopulations undetected by static MIC assays, and even evaluate the kinetics of heteroresistance and assess whether heteroresistant cells eventually adapt to drug pressure, which makes for wide clinical applicability [172,173,174]. In *C. neoformans*, time kill assays have been used to study azole and AmB heteroresistance, revealing that subpopulations of cells can survive for prolonged periods despite the presence of high drug concentrations [57]. These make time-kill assays a valuable tool in understanding heteroresistance dynamics.

### 8.3. Treatments of Antifungal Heteroresistance

There are currently no approved first-line therapies for heteroresistance. For bacteria, a combination of several antibiotics serves as the preferred strategy [28,174,175,176]. For example, combination polymyxin B and tigecycline therapy was shown to kill both the polymyxin B- and tigecycline-heteroresistant *Klebsiella pneumoniae* [174]. Similarly in *Acinetobacter baumannii*, combined tigecycline and colistin therapy was bactericidal on TGC and colistin multi-heteroresistance [175]. For fungi, combination of drugs or dosage escalation may aid in alleviating treatment failure. A study of *C. neoformans* showed that 5-FC effectively suppress fluconazole-induced heteroresistance, suggesting the importance of combined therapy [142]. Other research suggested the feasibility of fluconazole dose escalation and combination therapy in the treatment of cryptococcal meningitis but also revealed limitations in combination options, for which only 5-FC was widely available [165]. However, comparative studies and clinical trials remain limited. Expanding the scope to all failed antifungal treatments, such as adding 5-FC or amphotericin B to fluconazole monotherapy, alleviates mortality rate and achieves up to 90% efficacy [177,178,179]. Considering the evolution of resistance or cross-resistance, even suitable combination therapies or escalation dose therapies can increase the risk of toxicity and adverse effects. Inspired by well-studied combination antibiotic therapy and results from the limited use of combination antifungal therapy, addition of different classes of antifungals has the potential to clear heteroresistance. In-depth study of the mechanisms of heteroresistance may help develop novel drugs and providing low-toxicity combination therapies.

## 9. Conclusions

Heteroresistance is a reversible form of antifungal resistance that fluctuates in response to varying conditions, primarily driven by genome aneuploidy and changes in gene expression. Unlike bacterial resistance, fungal heteroresistance remains largely underexplored, with most research focusing on azole resistance. A unique characteristic of heteroresistance is its inducibility, with resistant phenotypes emerging in response to antifungal exposure or other factors, leading to reduced drug efficacy over time.

Molecularly, aneuploidy and CNVs play a crucial role, influencing mechanisms such as gene loss, gene amplification, modifications of antifungal-binding sites, and upregulation of efflux pumps. These genetic changes enhance fungal adaptability to antifungal pressure. One point worth exploring is that CNVs can be stably inherited and may occur in both resistance and heteroresistance, which means that they may contribute greatly to the evolution of antifungal resistance. Understanding the evolutionary and molecular mechanisms of heteroresistance is crucial for predicting and controlling clinical resistance.

Antifungal heteroresistance poses a significant clinical challenge, as traditional diagnostic techniques may mistakenly report it as susceptibility and lead to misuse of antifungals. To make things worse, it can develop during treatment and even evolve into stable and heritable resistance, leading to therapeutic failure and recurrent infections. While escalating drug dosages or using combination therapies may sometimes be effective, comparative clinical studies remain limited to date. Moreover, these approaches come with limited options and increased toxicity risks. Therefore, the development of novel antifungal agents is critical, and is expected to provide more therapeutic choices and combination possibilities. In the final analysis, gaining deeper insights into the specific mechanisms of heteroresistance could help to gain deeper and clearer insights into phenotype, thus aiding the development of innovative treatment strategies targeting multiple pathways, ultimately improving patient outcomes and reducing healthcare costs.

## Figures and Tables

**Figure 1 jof-11-00143-f001:**
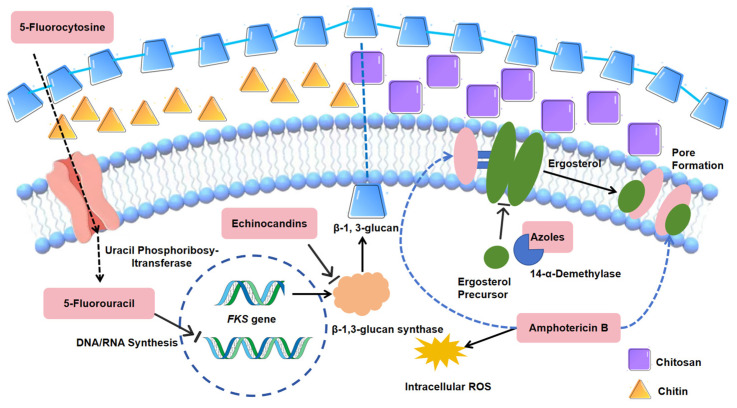
Antifungal mode of action of azoles, AMB, 5-FC and echinocandins. Azoles inhibit ergosterol biosynthesis by targeting sterol 14α-demethylase, producing toxic 14-methyl sterols, altering fungal cell membrane permeability and metabolic state [74,75]. Echinocandins target β-1,3-glucan synthase, impairing fungal cell wall integrity and stress resistance [76]. Amphotericin B binds to membrane ergosterol, forming pores that disrupt permeability and cause reactive oxygen species (ROS) accumulation, leading to cell death [77,78,79]. 5-FC is converted intracellularly to 5-FU, inhibiting DNA and RNA synthesis [80,81].

**Figure 2 jof-11-00143-f002:**
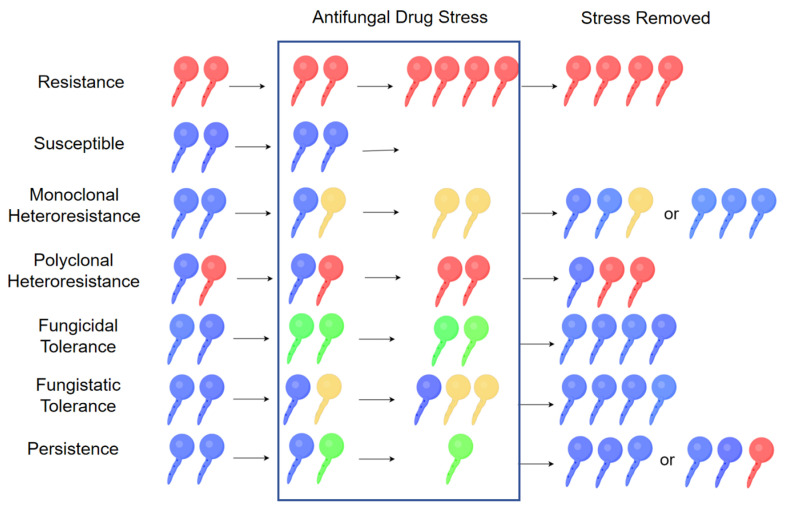
(By Figdraw 2.0). Explanation of resistance, susceptible, tolerance, persistence and heteroresistance from a cell population perspective. The state of cell proliferation is represented by the number of cells in the figure. Individual colors indicate different genotypes and phenotypes, which are red (genetically stable resistant), blue (susceptible), yellow (genetically unstable or phenotypically resistant), green (phenotypically tolerant).

**Table 1 jof-11-00143-t001:** Differences between resistance, tolerance, persistence and heteroresistance.

Concept	Description	Mechanisms	Clinical Features	Referrence
Resistance	The ability to withstand the effects of an antifungal drug whose concentration is typically effective.	Heritable mutations are associated with drug targets or efflux pumps expression, etc.	Treatment failure and increased risk of infection spread.	[82,83,84]
Tolerance	The whole population survives antifungal exposure at concentrations that would otherwise be lethal.	Altered cell growth or metabolism, etc.	High mortality rates, persistent infection and treatment failure.	[98,103,104,105,106,107,108,113]
Persistence	A subpopulation of genetically susceptible fungal cells survives fungicidal concentrations of drugs.	Altered cell growth and metabolism, etc.	Treatment failure and recurrence.	[109,110,111,112]
Heteroresistance	A minority of phenotypically resistant cells with elevated MICs coexist with the most susceptible cells.	Altered expression levels, aneuploidy and copy number variation, etc.	High mortality rates, treatment failure, potential to evolve into full resistance.	[15,18,48,57,99,100,101,102]

**Table 2 jof-11-00143-t002:** Heteroresistance Mechanisms in Common Fungal Pathogens.

Types	Species	Antifungals	Mechanisms	Related Components	References
Aneuploidy and CNVs	*C. albicans*	Fluconazole	Chr5 disomy	*ERG11* and TAC1	[129,130]
	*C. albicans*	5-Flucytosine	Loss of chr5, due to the location of negative regulator (s) of anti 5-FC resistance	Unknown.	[130]
	*C. albicans*	Fluconazole	“Trimeras”, three connected cells composed of a mother, daughter, and granddaughter bud	Unknown.	[131]
	*C. albicans*	Echinocandins	Chr2 trisomy	*RNR1*, *RNR21*	[132]
	*C. albicans*	Echinocandins (caspofungin, micafungin and anidulafungin)	Chr5 aneuploidy after caspofungin exposure can produce cross-resistance	Three negative regulators *CHT2*, *PGA4* and *CSU51*, and two positive regulators, *CNB1* and *MID1*.	[118,133]
	*N. glabrata*	Echinocandins (anidulafungin)	ChrE aneuploidy contributes to heteroresistance after exposing clinical isolates to anidulafungin.	Unknown.	[134]
	*N. glabrata*	Azoles	Incremental effects of multiple binary genetic switches	*CDR1*, *PDH1*, *PDR1* and *SNQ1*	[99]
	*N. glabrata*	Azoles	Formation of “trimeras”	Unknown.	[131]
	*C. parapsilosis*	Azoles	Formation of “trimeras”	Unknown.	[131]
	*C. auris*	Azoles (fluconazole)	Genome changes mainly involving SNP, with aneuploidy a minority. But due to its haploid genome, SNPs may have immediate phenotypic impact.	SNPs	[135,136,137]
	*C. gattii*	Azoles (fluconazole)	Copy number increase in chr10, chr9 and chr11 in VGI genotype and copy number increase in chr3 in VGII genotype	Unknown.	[138]
	*C. neoformans*	Cross-resistance to 5-FC and Fluconazole	Chr1 disomy	*ERG11*, *AFR1*	[139]
	*C. neoformans*	Fluconazole	Overexpression of *AFR1* on chr1 and *GEA2* on chr3	*AFR1*, *GEA2*	[139]
	*C. neoformans*	Azoles (fluconazole)	Titan cells that produce multiple types of aneuploid daughter cells	Unknown.	[56,140,141]
	*C. neoformans*	Azoles (fluconazole)	Chr1 disomy	*ERG11*, *AFR1*	[142]
	*C. neoformans*	Azoles (fluconazole)	Chr4 disomy	*SEY1*, *GCS2*, *GLO3*	[143]
	*C. neoformans*	Azoles (fluconazole)	Chr3 disomy caused by gene relocation	*ERG11*, *SEY1*, *GCS2*, *GLO3*	[143,144]
Alterations in Gene Expression	*C. albicans*	Azoles	Elevation of mRNA	ATP Binding Cassette superfamily *CDR* genes	[145]
	*C. gattii*	Azoles (fluconazole)	Up-regulated activity of efflux pumps	*PDR11*	[146]
	*C. neoformans*	Azoles (fluconazole)	Up-regulated activity of efflux pumps	*AFR1*	[142]
Environmental Stress	*C. neoformans*	Polyene (AMB) and Azoles (fluconazole)	Nitrogen limitation	Unknown.	[57]
	*C. neoformans*	Azoles (fluconazole)	Environmental temperature	Unknown.	[19]
	*C. neoformans*	Azoles	Temperature, media type, growth phase, and the age of cells	Unknown.	[147]

## Data Availability

This is a review article, and no new data were created or analyzed. All data discussed in this study are publicly available in the references cited.

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
