# Peer review of "Insight into the Mechanisms and Clinical Relevance of Antifungal Heteroresistance"

_jof, 2025, doi:10.3390/jof11020143_

Round 1
Reviewer 1 Report
Since the authors discuss the clinical relevance of antifungal heteroresistance, I suggest including more studies that investigate its effects in mice or other robust in vivo models. Currently, only one study is mentioned in the manuscript. Furthermore, I recommend incorporating data on antifungal resistance prevalence, mortality rates, and associated healthcare costs. This data would further emphasize the clinical impact of heteroresistance.
Introduction
- I suggest including data on the prevalence and mortality rates of fungal infections worldwide.
- The concept of antifungal resistance described (L56) is based on laboratory findings. Since the authors discuss the clinical relevance of heteroresistance, I suggest incorporating the clinical definition of heteroresistance.
- L74, L77 – Please italicize species names.
Topic 2
- I recommend modifying the title to reflect the discussion on antifungal resistance mechanisms, as this is also a focus of this section.
- Please add information about the fungicidal/fungistatic mode of action of antifungals, as this is relevant to heteroresistance events.
Minor points
- L118, L164, L185, Topic 6.1.2, L365, Table 1, L431 – Candida glabrata has been reclassified as Nakaseomyces glabrata. Please update this information throughout the manuscript.
- L133, L431 – Candida lusitaniae has been reclassified as Clavispora lusitaniae. Please update this information throughout the manuscript.
- L250 – Please correct the numbering in the title. The correct number is 6.1.
Author Response
- Reviewer 1
- Comments 1: I suggest including data on the prevalence and mortality rates of fungal infections worldwide.
Response 1: Thank you for pointing this out. I agree with this comment. Therefore, I have included specific data on the prevalence and mortality rates of fungal infections. This change can be found - page 3, part 1, and line 53-60.] - Comments 2: The concept of antifungal resistance described (L56) is based on laboratory findings. Since the authors discuss the clinical relevance of heteroresistance, I suggest incorporating the clinical definition of heteroresistance.
Response 2: Thank you for pointing this out. l totally agree with this. I have incorporated the clinical definition of heteroresistance. This change can be found - page 5, part 4, and line 194-195.] - Comments 3: L74, L77 – Please italicize species names.
Response 3: Thank you for pointing this out and please forgive my carelessness. I have checked and italicized species names throughout the whole manuscript. - Comments 4: L118, L164, L185, Topic 6.1.2, L365, Table 1, L431 – Candida glabratahas been reclassified as Nakaseomyces glabrata. Please update this information throughout the manuscript. L133, L431 – Candida lusitaniae has been reclassified as Clavispora lusitaniae. Please update this information throughout the manuscript.
Response 4: Thank you for pointing this out. I have updated the classification of Nakaseomyces glabrata and Clavispora lusitaniae throughout the whole manuscript. - Comments 5: I suggest including more studies that investigate its effects in mice or other robust in vivo
Response 5: Thank you for pointing this out. I agree with this comment and added a few researches. However, there are relatively very few in vivo studies to date, so I would appreciate it if you understand that there's not much to add. The change can be found - page 8, part 8.1, and line 444-457.] - Comment 6: From the review report “The discussion must be improved.”
Response 6: Thank you for pointing this out. I agree with this comment. I have improved my writing in the final part “conclusion”. This change can be found in page 14, part 9, line 548-559. - Comment 7: Topic 2- I recommend modifying the title to reflect the discussion on antifungal resistance mechanisms, as this is also a focus of this section.
- Please add information about the fungicidal/fungistatic mode of action of antifungals, as this is relevant to heteroresistance events.
Response: Thank you for pointing this out. I totally agree with this. I have changed the name of part 3, which can be seen in page 3, Line 111. Also, I illustrated a picture to give a detailed explanation of mode of action of antifungals, which can be seen in Fig 1.
- Comments 1: I suggest including data on the prevalence and mortality rates of fungal infections worldwide.
Reviewer 2 Report
This review addresses a critical gap in understanding antifungal heteroresistance, a phenomenon less studied in fungi compared to bacteria. While traditional mechanisms of antifungal resistance (e.g., stable mutations, efflux pumps) are well characterized, heteroresistance characterized by antifungal resistance is poorly understood despite its clinical relevance. The authors summarize the available fragmentary studies into a coherent framework, emphasizing mechanisms such as aneuploidy, copy number variations (CNVs), and stress-induced adaptations. By highlighting the differences in the terms tolerance and persistence, the review clarifies terminology that often overlaps in the existing literature. The focus on clinical consequences (e.g., treatment failure, relapse) is consistent with global concerns about antifungal resistance and highlights the urgency of addressing this adaptive survival strategy.
A review article was recently published - Beyond resistance: antifungal heteroresistance and antifungal tolerance in fungal pathogens Current Opinion in Microbiology Volume 78, April 2024, 102439. Authors are encouraged to read it and note the main differences from their own work, if possible.
L.92: The introduction provides references to 20 sources after one summary sentence, I think it is worth giving more details and referring more specifically.
L.266: the word fluconazole using twice, please check.
Author Response
- Reviewer 2
- Comments 1: The introduction provides references to 20 sources after one summary sentence, I think it is worth giving more details and referring more specifically.
Response 1: Thank you for pointing this out. I agree with this comment. Therefore, I have given more details and referred specifically according to each articles' main idea. This change can be found - page 3, part 3, and line 111-116.] - Comments 2: L.266: the word fluconazole using twice, please check.
Response 2: Thank you for pointing this out. I have corrected my spelling. This change can be found - page 9, part 7.1, and line 295.] - Comment 3: A review article was recently published - Beyond resistance: antifungal heteroresistance and antifungal tolerance in fungal pathogens Current Opinion in Microbiology Volume 78, April 2024, 102439. Authors are encouraged to read it and note the main differences from their own work, if possible.
Response 3: Thanks for your recommending. I’ve read this review article and found it a remarkable contribution to the field of antifungal research. It provides clear definitions and insights into antifungal resistance, heteroresistance, and tolerance, highlighting their clinical significance. The authors' emphasis on the need for standardized assays and consensus definitions is particularly valuable, making this work essential reading for both researchers and clinicians. I’d like to point out the main differences between these two reviews. "Beyond resistance: antifungal heteroresistance and antifungal tolerance in fungal pathogens," offers a comparative review, emphasizing the need for standardized definitions and assays across different research communities. It broadly covers various fungal pathogens and antifungal agents, aiming to unify understanding and approaches to these phenomena. In contrast, our review provides a detailed exploration of the specific mechanisms underlying antifungal heteroresistance, such as aneuploidy and gene expression changes, and delves into the clinical implications, including diagnostic challenges and potential therapeutic strategies. And our review is more focused on detailed experimental methods and clinical relevance. In summery, Yang’s review is more conceptual and comparative, whereas ours is more mechanistic and clinically oriented.
- Comments 1: The introduction provides references to 20 sources after one summary sentence, I think it is worth giving more details and referring more specifically.
Reviewer 3 Report
The article addresses antifungal heteroresistance, a poorly explored phenomenon that represents a significant challenge in antifungal therapy. Its clinical relevance is unquestionable, as heteroresistance can contribute to therapeutic failure and the development of stable resistance in medically important fungal pathogens.
I recommend minor revision:
- Include a methodological section describing how the reviewed studies were selected.
- Present a comparative table of the differences between heteroresistance, tolerance, and persistence.
- Expand the discussion on the impact of heteroresistance on diagnostic tests.
- Add information on therapeutic strategies and experimental data on antifungal combinations.
- Since this is a review, an experimental methodology is not expected, but a more detailed explanation of the criteria for selecting reviewed studies would be beneficial.
- Lack of clear differentiation between heteroresistance, tolerance, and persistence. Although these three categories are addressed, the explanation could be more precise and supported with concrete examples.
- Limited discussion on variability in antifungal susceptibility testing. The article does not deeply address how heteroresistance could influence the interpretation of sensitivity tests in clinical practice.
- Lack of a more developed therapeutic perspective. Although the combination of antifungals is mentioned as a strategy, specific data on their efficacy in clinical studies are lacking.
Author Response
Reviewer 3
- Comments 1: Include a methodological section describing how the reviewed studies were selected.
Response 1: Thank you for pointing this out. I agree with this comment. Therefore, I have added a methodological section. Please see page 3, part 2 Methodology. - Comments 2: Present a comparative table of the differences between heteroresistance, tolerance, and persistence.
Response 2: Thank you for pointing this out. I agree with this comment. Therefore, I have added a table. Please see page 6, Table 1. - Comments 3: Expand the discussion on the impact of heteroresistance on diagnostic tests.
Response 3: Thank you for pointing this out. I agree with this. Therefore, I have expanded the discussion on the impact of heteroresistance on diagnostic tests. This change can be found - page 12, part 8.2, and line 475-487.] - Comments 4: Lack of a more developed therapeutic perspective. Although the combination of antifungals is mentioned as a strategy, specific data on their efficacy in clinical studies are lacking.
Response 4: Thank you for pointing this out. However, there's still relatively limited studies in antifungal heteroresistance to date, so I expanded the scope of alternative therapy to bacterial heteroresistance and other kinds of antifungal treatment failure. The revised manuscript this change can be found - page 13, part 8.3, and line 512-532.] I would appreciate it if you could understand such difficulties. - Comments 5: Limited discussion on variability in antifungal susceptibility testing. The article does not deeply address how heteroresistance could influence the interpretation of sensitivity tests in clinical practice.
Response 5: Thank you for pointing this out. l totally agree with this. Therefore, I have expanded the discussion on how heteroresistance could influence the interpretation of sensitivity tests in clinical practice. This change can be found - page 14, part 9, and line 548-559.] - Comment 6: Lack of clear differentiation between heteroresistance, tolerance, and persistence. Although these three categories are addressed, the explanation could be more precise and supported with concrete examples.
Response 6: Thank you for pointing this out. I agree with this comment. Therefore, I have added a table. Please see page 6, Table 1.
Round 2
Reviewer 1 Report
Not applicable
Not applicable
Author Response
Dear Reviewer:
We would like to express our heartfelt thanks to you for sparing no effort in revising our manuscript.
In this version, we have made minor revisions that primarily include the following aspects:
- We have polished the language to improve the manuscript's fluency and readability.
- We have revised and deleted a few ambiguous points to ensure clarity.
- We have ensured the correct formatting of the manuscript
- We have checked and ensured the correctness and reasonableness of each reference citation.
- We have adjusted the position of Table 2 according to the content of the review.
All modified areas have been marked in red font. Please review them!